# Cardiovascular risk factors and outcomes in COVID-19: A hospital-based study in India

**Arvind K. Sharma** [1], **Vaseem Naheed Baig** [1], **Sonali Sharma** [2], **Gaurav Dalela**[3], **Raja Babu Panwar**[4], **Vishwa Mohan Katoch** [4], **Rajeev Gupta** [4,5]*

**1** Departments of Community Medicine, Biochemistry, Jaipur, India, **2** Departments of Community Medicine, Microbiology, Jaipur, India, **3** RUHS College of Medical Sciences, Rajasthan University of Health Sciences, Jaipur, India, **4** Academic Research Development Unit, Rajasthan University of Health Sciences, Jaipur, India, **5** Department of Medicine, Eternal Heart Care Centre & Research Institute, Jaipur, India

* rajeevgg@gmail.com, drrajeev.gupta@eternalheart.org

**Data Availability Statement:** All data are in the manuscript and supporting information files.

**Funding:** The authors received no specific funding for this work.

## Abstract

### Background & objectives

Presence of cardiovascular (CV) risk factors enhance adverse outcomes in COVID-19. To determine association of risk factors with clinical outcomes in India we performed a study.

### Methods

Successive virologically confirmed adult patients of COVID-19 at a government hospital were recruited at admission and data on clinical presentation and in-hospital outcomes were obtained. The cohort was classified according to age, sex, hypertension, diabetes and tobacco use. In-hospital death was the primary outcome. Logistic regression was performed to compared outcomes in different groups.

### Results

From April to September 2020 we recruited 4645 (men 3386, women 1259) out of 5103 virologically confirmed COVID-19 patients (91.0%). Mean age was 46±18y, hypertension was in 17.8%, diabetes in 16.6% and any tobacco-use in 29.5%. Duration of hospital stay was 6.8±3.7 days, supplemental oxygen was in 18.4%, non-invasive ventilation in 7.1%, mechanical ventilation in 3.6% and 7.3% died. Unadjusted and age-sex adjusted odds ratio (OR) and 95% confidence intervals(CI) for in-hospital mortality, respectively, were: age ≥60y vs <40y, OR 8.47(95% CI 5.87–12.21) and 8.49(5.88–12.25), age 40-59y vs <40y 3.69(2.53–5.38) and 3.66(2.50–5.33), men vs women 1.88(1.41–2.51) and 1.26(0.91–1.48); hypertension 2.22(1.74–2.83) and 1.32(1.02–1.70), diabetes 1.88(1.46–2.43) and 1.16(0.89–1.52); and tobacco 1.29(1.02–1.63) and 1.28(1.00–1.63). Need for invasive and non-invasive ventilation was greater among patients in age-groups 40–49 and ≥60y and hypertension. Multivariate adjustment for social factors, clinical features and biochemical tests attenuated significance of all risk factors.

**Competing interests:** The authors have declared that no competing interests exist.

## Conclusion

Cardiovascular risk factors, age, male sex, hypertension, diabetes and tobacco-use, are associated with greater risk of in-hospital death among COVID-19 patients.

## Introduction

Presence of cardiovascular risk factors (smoking, diabetes, hypertension, sedentary lifestyle and obesity, etc.) and clinical cardiovascular disease are associated with adverse outcomes in COVID-19 [1]. It has been reported that these factors lead to rapid progression of clinical manifestations, more severe pulmonary disease, greater requirement for oxygen and ventilatory support and greater mortality [1, 2]. It has also been reported that presence of hypertension, diabetes and cardiovascular disease is associated with a two-fold increase in risk of death in COVID-19 [2]. In a meta-analysis of 109 studies and 20,296 patients, the risk of mortality was higher in patients with increasing age, male sex (relative risk, RR 1.45, 95% confidence intervals, CI, 1.23–1.71), diabetes (1.59, 1.41–1.78), hypertension, tobacco use and congestive heart failure (4.76, 1.34–16.97) [3]. In another meta-analysis of 45 studies with 18,300 patients a significant association of in-hospital death was observed with age (coefficient 1.06, 95% CI 1.04–1.09), diabetes (1.04, 1.02–1.07) and hypertension (1.01, 1.01–1.03), but remained significant only for diabetes after statistical adjustment [4]. A third meta-analysis of 51 studies and 48,317 patients, mostly from high and upper middle income countries, reported relative risk of developing severe disease or deaths as significantly higher in patients with hypertension (RR 2.50, 95% CI 2.15–2.90), diabetes (2.25, 1.89–2.69) and cardiovascular disease (3.11, 2.55–3.79); the risk being significantly greater in older than younger individuals [5]. Population-based studies have identified importance of cardiovascular risk factors and disease in COVID-19 incidence and outcomes [6, 7].

Burden of COVID-19 related mortality is high in lower-middle and low-income countries of Asia and Africa [8]. Prevalence of cardiovascular risk factors is high in these countries.[9] India has high burden of cardiovascular risk factors [9], and COVID-19 cases and deaths [10]. There are limited data on association of cardiovascular risk factors with disease incidence and outcomes in India [11–13]. A macrolevel study in India reported that states with higher prevalence of cardiovascular risk factors- aging, hypertension, diabetes and obesity- had significantly higher COVID-19 incidence and deaths [14]. Although higher age is well-known COVID-19 risk factor, controversy exists regarding relative importance of risk factors- hypertension, diabetes or tobacco use [2]. Therefore, to evaluate association of cardiovascular risk factors (age, male sex, hypertension, diabetes and any tobacco use) in virologically confirmed COVID-19 cases successively admitted to a government hospital in India we performed a registry-based study.

## Methods

We conducted a hospital based observational study on patients with laboratory confirmed COVID-19 admitted to a 1200-bed dedicated COVID-19 government hospital (Rajasthan University of Health Sciences Hospital, Jaipur, India) from April to mid-September 2020. Initial data on some of these patients have been reported earlier [15–17]. The registry has been approved by the college administration and institutional ethics committee (CDSCO Registration Number: CR/762/Inst/RJ/2015). It is registered with Clinical Trials Registry of India at

www.ctri.nic.in with number REF/2020/06/034036. Individual patient consent was waivered by the ethics committee and anonymized data have been used with no patient identifiers.

## Patient data

Successive adult patients (more than 18 years) presenting to the hospital for admission with suspicion of COVID-19 infection were enrolled in the study and those who tested positive for COVID-19 on nasopharyngeal and oropharyngeal reverse transcriptase-polymerase chain reaction (RT-PCR) test were included. Details of methodology have been reported [16]. All RT-PCR positive patients admitted from 1 April to 15 September have been included. Patients recruited into the study in mid-September were followed up to discharge or death and outcome events were recorded.

A questionnaire was developed and details of sociodemographic, clinical, laboratory, treatments and outcomes variables were recorded using patient-history and medical files. Demographic details were obtained at the time of admission. These included name, age, sex, residence address, educational status. History of tobacco use (smoking, smokeless tobacco) and past hypertension, diabetes, cardiovascular diseases and other chronic diseases was also recorded at admission. Current smokers and users of smokeless tobacco were categorized as tobacco use. Hypertension and diabetes were diagnosed from history of known disease or on treatment. Details of physical examination at the time of admission were obtained from patient case files. These included history of duration of symptoms at admission, pulse, blood pressure (BP), respiratory rate, and surface oxygen concentration (SpO2) using digital devices. We could not obtain details of body mass index as height and weight was not routinely recorded on admission. Details of investigations at admission were obtained from the case files and biochemistry, microbiology and pathology departments as reported earlier [15, 16]. We do not have data on serial investigations. We obtained data on duration of hospital stay from medical record department. For patients discharged alive from the hospital, we obtained data on number of patients who needed oxygen support (nasal prongs, facial mask or high-flow nasal cannula), non-invasive ventilation (CPaP or BiPaP support) or invasive ventilation after endotracheal intubation. Binary outcomes were obtained for all patients and included recovery, referral to non-government hospitals on request of family, or death. In-hospital death was the primary outcome while requirement for invasive and non-invasive ventilation were secondary outcomes. All these data are being routinely sent to the Department of Health, Government of Rajasthan, India, but are not currently accessible.

## Statistical analyses

The data were computerized and processing was performed using commercially available statistical software, SPSS v.20.0. Numerical data are expressed as numbers ±1 SD and categorical data as percent. Significance of intergroup differences were calculated using unpaired t-test or ANOVA for continuous variables and $\chi^2$ test for categorical variables. To evaluate association of COVID-19 related in-hospital deaths and other adverse outcomes (invasive ventilation, non-invasive ventilation) with age, male sex, hypertension, diabetes and tobacco use, we performed stepwise logistic regression. Univariate and multivariate odds ratios (OR) and 95% confidence intervals (95% CI) were calculated. In the first step we calculated univariate odds ratio. Age- and sex-adjusted odds ratios were calculated in the second step. For multivariate adjusted odds ratios we added household size, educational status, comorbidities, risk factors (other than the risk factor in question) and clinical severity (oxygen level at admission, need for oxygenation) and calculated OR and 95% CI. P value <0.05 is considered significant.

## Results

Data were obtained from April to mid-September 2020. During this period, a total of 7349 patients were hospitalized with confirmed or suspected COVID-19, 5103 patients (69.0%) tested positive for the disease on RT-PCR test and for the present study 4645 adult men and women ≥18 years (91.0% of confirmed cases), men 3386 (72.9%) and women 1259 (27.1%), in whom detailed clinical data were available have been included. The mean age of the cohort was 46±18 years, 54% were less than 50 years and about half lived in large family households (>5 persons). Prevalence of low educational status was higher in women while tobacco use was more in men. Comorbidities were present in 28.6% with hypertension (17.8%) and diabetes (16.6%) being the most common. Other comorbidities were chronic pulmonary disease, tuberculosis, coronary heart disease and neurological disease (Fig 1). Data on hematological investigations were available in 4456 (95.9%) and for biochemical tests in 867 (18.7%). All patients received standard treatment according to guidelines available from Indian Council of Medical Research and the State government [18]. The average length of stay in hospital was 6.8 ±3.7 days. Oxygen requirement was in 861 (18.4%), non-invasive ventilation or high flow oxygen in 334 (7.1%) and mechanical ventilation in 169 (3.6%). In-hospital mortality was in 340 patients (7.3%).

Clinical characteristics, important clinical findings, selected investigations and outcomes in patients aged >40 years, 40–59 years and ≥60 years and are shown in Table 1. Older participants were less educated, lived in larger households and had greater prevalence of tobacco use, hypertension, diabetes, and cardiovascular disease with significantly higher systolic blood pressure (BP) and more hypoxia (Table 1). There were no differences in hematological and biochemical parameters. Oxygen requirement, non-invasive as well as invasive ventilation were more in older age-groups (40–59 and ≥60 years) with graded escalation. Deaths were significantly higher in age-group 40–59 (7.1%) and ≥60 (15.0%) when compared to <40 years (2.1%) (Fig 2). Table 2 shows that women were less literate and had lower prevalence of hypertension, diabetes and tobacco use. Oxygen requirement was significantly more in women while requirement of non-invasive or invasive ventilation were not different. Hematological and biochemical parameters were not significantly different and are not shown. Number of in-hospital deaths were significantly more in men (n = 282, 8.3%) as compared to women (n = 58, 4.6%) (Fig 2) with univariate OR 1.88, (95% CI 1.41–3.51).

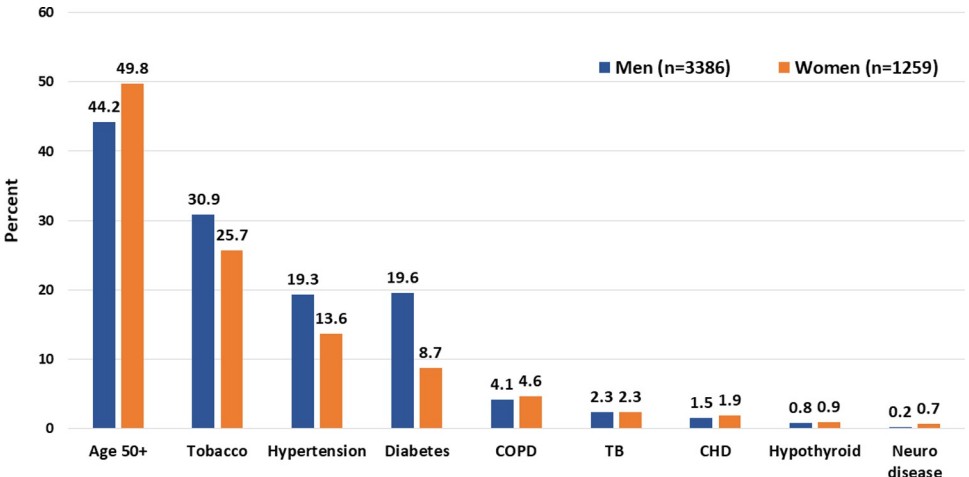

**Fig 1. Distribution of cardiovascular risk factors among men and women in the study cohort.**

**Table 1. Clinical characteristics and outcomes in age-groups <40 years, 40–59 years and ≥60 years.**

| | Age 40 yrs (N = 1757) | Age 40–59 yrs (N = 1716) | Age ≥60 (N = 1203) | p-value* |
|---|---|---|---|---|
| Age: mean±SD | 26.5±8.1 | 49.5±5.7 | 68.4±7.1 | <0.001 |
| Low Educational Status | 395(37.2) | 497(48.7) | 532(60.5) | <0.001 |
| Household members: mean±SD | 3.4±2.2 | 3.6±1.9 | 4.0±2.7 | <0.001 |
| **Risk factors** | | | | |
| Smoking/tobacco | 480(27.3) | 512(29.8) | 377(31.3) | 0.050 |
| Hypertension | 34(1.9) | 392(22.8) | 405(33.7) | <0.001 |
| Type 2 Diabetes | 52(3.0) | 357(20.8) | 368(30.6) | <0.001 |
| Thyroid disease | 5(0.3) | 21(1.2) | 12(1.0) | 0.006 |
| Coronary heart disease | 7(0.4) | 25(1.5) | 43(3.6) | <0.001 |
| Chronic pulmonary disease | 99(5.6) | 66(3.8) | 28(2.3) | <0.001 |
| **Clinical findings** | | | | |
| Systolic BP mmHg | 121.0±7.4 | 126.3±12.5 | 129.6±13.2 | <0.001 |
| SpO2<90% | 81(4.6) | 158(9.2) | 118(9.8) | <0.001 |
| SpO2 90–94% | 173(9.8) | 208(12.1) | 180(15.0) | <0.001 |
| **Investigations** | | | | |
| Haemoglobin, g/dl | 12.8±2.2 | 12.7±2.4 | 12.5±2.2 | 0.062 |
| White cells, $10^9$ cells/L | 7.6±3.6 | 7.5±4.0 | 7.4±3.9 | 0.276 |
| Lymphocyte:Neutrophil Ratio | 0.36±39 | 0.35±79 | 0.36±81 | 0.348 |
| Sodium, mEq/L | 134.8±17.2 | 136.2±10.3 | 137.3±8.4 | 0.158 |
| Creatinine, mg/dl | 0.95±0.48 | 0.90±0.35 | 1.01±0.75 | 0.123 |
| **Clinical Outcomes** | | | | |
| Oxygen requirement | 222(12.6) | 314(18.3) | 325(27.0) | <0.001 |
| Non-invasive ventilation | 66(3.8) | 134(7.8) | 134(11.1) | <0.001 |
| Invasive ventilation | 24(1.4) | 68(4.0) | 77(6.4) | <0.001 |
| **In-hospital outcomes** | | | | |
| Recovered | 1757(97.0) | 1580(92.1) | 1010(84.0) | <0.001 |
| Referred | 16(0.9) | 13(0.8) | 12(1.0) | 0.777 |
| Deaths | 36(2.1) | 123(7.1) | 181(15.0) | <0.001 |

Numbers in parentheses are percent; Numbers ± are 1 SD

*p-value calculated using $\chi^2$-square test for categorical variables and ANOVA for continuous variables.

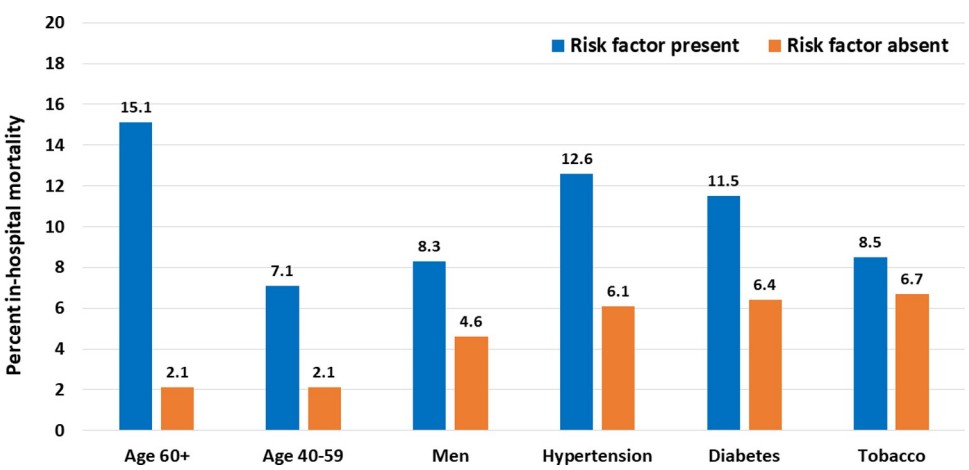

**Fig 2. In-hospital deaths in COVID-19 patients with or without the risk factor.**

**Table 2. Clinical characteristics and outcomes according and to sex.**

|  | Men (n = 3386) | Women (n = 1259) | Univariate Odds Ratio (95% CI) |
|---|---|---|---|
| Age: mean±SD | 45.5±17.8 | 47.1±18.5 | 1.60[0.43–2.76] |
| Low educational status | 800[23.4] | 624[49.8] | 0.31[0.27–0.36]*** |
| **Risk factors** | | | |
| Smoking/tobacco | 1086[31.9] | 283[22.3] | 1.64[1.41–1.89]*** |
| Hypertension | 658[19.3] | 173[13.6] | 1.51[1.26–1.81]*** |
| Type 2 Diabetes | 666[19.6] | 111[8.7] | 2.53[2.05–3.13]*** |
| Thyroid disease | 27[0.8] | 11[0.9] | 0.91[0.45–1.84] |
| Coronary heart disease | 51[1.5] | 24[1.9] | 0.79[0.48–1.28] |
| Chronic pulmonary disease | 135[4.0] | 58[4.6] | 0.85[0.63–1.18] |
| **Clinical findings** | | | |
| Systolic BP mmHg | 125.1±11.9 | 126.0±12.9 | 0.88[0.09–1.66]* |
| SpO2 <90% | 252[11.4] | 105[12.2] | 0.88[0.69–1.12] |
| SpO2 90–94% | 397[18.0] | 164[19.1] | 0.89[0.73–1.08] |
| **Clinical Outcomes** | | | |
| Oxygen requirement | 600[17.6] | 261[20.6] | 0.77[0.65–0.90]** |
| Non-invasive ventilation | 236[6.9] | 98[7.7] | 0.88[0.69–1.33] |
| Invasive ventilation | 123[3.6] | 46[3.6] | 0.99[0.70–1.40] |
| **In-hospital outcomes** | | | |
| Recovered | 3020[88.7] | 1197[94.3] | 0.95[0.39–0.65]*** |
| Referred | 104[3.0] | 15[1.2] | 0.77[0.54–1.09] |
| Deaths | 282[8.3] | 58[4.6] | 1.88[1.41–2.51]*** |

Odds ratio for categoric al variables; Mean difference for continuous variables; CI confidence intervals

***p<0.001

**p<0.01

*p<0.05.

Risk factors, clinical findings and outcomes in patients with hypertension, diabetes and smoking/tobacco use are shown in Tables 3–5. Hematological and biochemical parameters did not show significant inter-group differences (data not shown). Patients with known hypertension were older and had higher prevalence of diabetes, cardiovascular disease, hypothyroidism and smoking/tobacco (Table 3). Need for non-invasive ventilation was also more in hypertension (OR 1.82, CI 1.41–2.35). Deaths were significantly greater in patients with hypertension (12.6%) versus in those without hypertension (6.1%) (Fig 2) (univariate OR 2.22, CI 1.74–2.83). Patients with diabetes were older with greater prevalence of smoking/tobacco, hypertension, pulmonary disease and cardiovascular disease with higher admission BP (Table 4). As compared to non-diabetics, number of deaths were significantly greater in patients with diabetes (n = 89, 11.5%) versus without diabetes (n = 251, 6.4%) (Fig 2) (univariate OR 1.88, CI 1.46–2.43). Smoking/tobacco-user group (smokers, smokeless tobacco) had more men, with greater prevalence of hypertension, diabetes and cardiovascular disease (Table 5). Compared with non-tobacco users the need for oxygen (OR 1.23, CI 1.05–1.44) and non-invasive ventilation (OR 1.31, CI 1.04–1.66). Deaths were significantly among tobacco users (n = 117 (8.5%) as compared to non-tobacco users (n = 223, 6.7%) (Fig 2) with OR 1.29 (95% CI 1.02–1.63).

We also performed age-sex adjusted and multivariate analyses to determine association of various cardiovascular risk factors with COVID-19 related in-hospital mortality (Table 6) and other outcomes (Table 7). Age ≥60 years vs <40 years emerged as the most important risk factor with significantly greater deaths on univariate (OR 8.47, 95% CI 5.87–12.21), sex-adjusted

**Table 3. Clinical characteristics and outcomes according to hypertension.**

| Variables | Hypertension N = 831 | Non-hypertension N = 3845 | Univariate Odds Ratio (95% CI) |
|---|---|---|---|
| Mean age | 58.5±12.4 | 43.0±18.0 | -15.5[-16.8–14.2]*** |
| Men | 658[19.3] | 2748[80.7] | 1.52[1.26–1.82]** |
| Women | 173[13.6] | 1097[86.4] | |
| Low educational status | 248[29.8] | 1176[31.0] | 0.95[0.88–1.14] |
| **Risk factors** | | | |
| Smoking/tobacco | 345[41.5] | 1024[26.6] | 1.95[1.67–2.28]** |
| Type 2 Diabetes | 515[61.9] | 316[8.1] | 22.3[18.5–27.0]*** |
| Thyroid disease | 12[1.4] | 316[8.2] | 2.15[1.08–4.28]* |
| Coronary heart disease | 32[3.9] | 26[0.7] | 3.54[2.23–5.63]*** |
| Chronic pulmonary disease | 34[4.1] | 159[4.1] | 0.99[0.68–1.44] |
| **Clinical findings** | | | |
| Systolic BP mmHg | 137.7±13.1 | 122.4±9.2 | -15.3[-16.1–14.6]*** |
| SpO2 <90% | 100[12.5] | 386[10.4] | 1.22[0.97–1.55] |
| SpO2 90–94% | 132[16.4] | 569[15.3] | 1.09[0.88–1.34] |
| **Clinical Outcomes** | | | |
| Oxygen requirement | 171[19.9] | 660[17.3] | 1.18[0.98–1.43] |
| Non-invasive ventilation | 91[11.0] | 243[6.3] | 1.82[1.41–2.35]** |
| Invasive ventilation | 31[18.3] | 800[17.3] | 1.04[0.70–1.55] |
| **In-hospital outcomes** | | | |
| Recovered | 697[83.9] | 3520[91.5] | 0.33[0.26–0.41]*** |
| Referred | 29[3.5] | 90[2.4] | 1.51[0.98–2.31] |
| Deaths | 105[12.6]] | 235[6.1] | 2.22[1.74–2.83]*** |

Odds ratio for categoric al variables; Mean difference for continuous variables; CI confidence intervals

***p<0.001

**p<0.01

*p<0.05.

(OR 8.49, CI 5.88–12.25) and multivariate analyses (OR 7.25, CI 4.92–10.66). In age-group 40–59 years also deaths were significantly higher than <40 y on univariate and multivariate analyses (Table 6). On univariate analyses (Table 6), male sex (OR 1.88, 1.41–2.51), hypertension (2.22, 1.74–2.83), diabetes (1.88, 1.46–2.43) and tobacco (1.29, 1.02–1.63) were associated with significantly more deaths (p<0.001) (Fig 3). There was moderate attenuation of significance with age and sex adjusted analyses, but hypertension (1.32, 1.02–1.70) and tobacco use (1.28, 1.00–1.63) continued to be significant. Following multivariate analyses significance of all the risk factors completely attenuated (Fig 3). Analyses of secondary outcomes show that in patients age ≥60 years as well in age-group 40–59 years, compared with <40 years, the need for invasive ventilation as well as non-invasive ventilation were higher (Table 7). Hypertension was significantly associated with greater risk of invasive ventilation on univariate and adjusted analyses and greater risk of non-invasive ventilation on univariate analyses. Diabetes patients had greater risk of non-invasive and invasive ventilation on univariate analyses which attenuated on age-sex adjusted and multivariate analyses (Table 7).

## Discussion

This study shows that multiple cardiovascular risk factors- hypertension, diabetes, tobacco use, increasing age and male sex are associated with greater risk of death and adverse outcomes among hospitalized COVID-19 patients on univariate analyses. Age is the most important

**Table 4. Clinical characteristics and outcomes according to diabetes.**

| Variables | Diabetes N = 777 | Non-Diabetes N = 3899 | Univariate Odds Ratio (95% CI) |
|---|---|---|---|
| Mean age | 57.6±12.9 | 43.4±18.1 | -14.2[-15.5–12.9]*** |
| Men | 549[16.1] | 2857[83.9] | 0.53[0.71–1.00]* |
| Women | 228[18.0] | 1042[82.0] | |
| Low educational status | 220[28.3] | 1204[31.3] | 0.87[0.74–1.03] |
| **Risk factors** | | | |
| Smoking/tobacco | 303[39.0] | 1066[27.4] | 1.70[1.45–1.99]** |
| Hypertension | 515[66.3] | 316[8.1] | 22.3[18.5–27.0]*** |
| Thyroid disease | 10[1.3] | 28[0.7] | 1.80[0.87–3.73] |
| Coronary heart disease | 27[3.5] | 48[1.2] | 2.89[1.79–4.66]*** |
| Chronic pulmonary disease | 48[6.2] | 145[3.7] | 1.70[1.22–2.38]** |
| **Clinical findings** | | | |
| Systolic BP mmHg | 135.1±13.7 | 123.1±10.0 | -11.9[-12.8–11.1]*** |
| SpO2 <90% | 92[11.84] | 394[10.10] | 1.19[0.94–1.52] |
| SpO2 90–94% | 119[15.31] | 582[14.92] | 1.03[0.83–1.28] |
| **Clinical Outcomes** | | | |
| Oxygen requirement | 187[24.1] | 674[17.3] | 1.52[1.26–1.82]*** |
| Non-invasive ventilation | 64[8.2] | 270[6.9] | 1.21[0.91–1.60] |
| Invasive ventilation | 41[5.3] | 128[3.3] | 1.64[1.14–2.35]** |
| **In-hospital outcomes** | | | |
| Recovered | 653[84.0] | 3564[91.4] | 0.49[0.39–0.62]*** |
| Referred | 35[4.5] | 84[2.2] | 2.14[1.43–3.20]*** |
| Deaths | 89[11.5] | 251[6.4] | 1.88[1.46–2.43]*** |

Odds ratio for categoric al variables; Mean difference for continuous variables; CI confidence intervals

***p<0.001

**p<0.01

*p<0.05.

predictor and there is graded increment of deaths and other adverse outcomes with increasing age. Significance of hypertension and tobacco-use is retained even after age and sex adjustment highlighting greater importance of these factors.

Our results are similar to most of the previous meta-analyses that have identified age as the most important risk factor for adverse outcomes in COVID-19 [2–5]. Cardiovascular risk factors- hypertension and diabetes- have been identified as important in many previous studies [3–5]. In the present study, although both of these factors are associated with greater risk of death and some adverse outcomes (Table 6), there is substantial attenuation after age-adjustment for hypertension and complete attenuation in diabetes. These findings indicate that age is important intermediate pathway of increased risk-factor associated mortality in COVID-19. A study limitation is that we used self-reported presence of hypertension and diabetes at the time of admission as risk factor. Given the fact that in India only about half of the patients with hypertension and two-thirds with diabetes are aware of their condition [19], the prevalence of these conditions might have been higher in our cohort. However, many individuals with hypertension present with low BP in acute COVID-19 and therefore estimation of prevalence of hypertension based on measured BP would have been erroneous. Moreover, our unadjusted OR of 2.22 (CI 1.74–2.51) and age-sex adjusted OR of 1.32 (CI 1.02–1.70) is similar to many previous studies and meta-analyses have calculated hypertension related OR in COVID-19 between 1.90 (CI 1.69–2.35) [3] and 2.50 (CI 2.15–2.90) [5], similar to the present study. We

**Table 5. Clinical characteristics and outcomes according to smoking/tobacco use.**

| Variables | Smoking/tobacco N = 1369 | Non-smoker/tobacco N = 3307 | Univariate Odds Ratio (95% CI) |
|---|---|---|---|
| Mean age | 46.3±18.1 | 45.5±18.1 | -0.86[-2.00–0.28] |
| Men | 1086[79.3] | 2320[70.2] | 1.64[1.41–1.89]*** |
| Women | 283[20.7] | 987[29.8] | |
| Low educational status | 414[30.6] | 1010[30.8] | 1.01[0.88–1.16] |
| **Risk factors** | | | |
| Hypertension | 345[25.2] | 486[14.7] | 1.96[1.67–2.28]*** |
| Type 2 Diabetes | 303[22.1] | 474[14.3] | 1.69[1.45–1.99]*** |
| Thyroid disease | 5[0.4] | 33[1.0] | 0.36[0.14–0.93]* |
| Coronary heart disease | 34[2.5] | 41[1.2] | 2.03[1.28–3.21]** |
| Chronic pulmonary disease | 67[4.9] | 126[3.8] | 1.29[0.96–1.76] |
| **Clinical findings** | | | |
| Systolic BP mmHg | 126.1±12.6 | 124.7±11.2] | -1.3[-2.1–0.6]*** |
| SpO2 <90% | 131[9.9] | 355[11.1] | 0.88[0.71–1.08] |
| SpO2 90–94% | 194[14.7] | 507[15.8] | 0.91[0.76–1.09] |
| **Clinical Outcomes** | | | |
| Oxygen requirement | 283[20.7] | 578[17.5] | 1.23[1.05–1.44]* |
| Non-invasive ventilation | 116[8.5] | 218[6.6] | 1.31[1.04–1.66]* |
| Invasive ventilation | 46[3.4] | 123[3.7] | 0.90[0.64–1.27] |
| **In-hospital outcomes** | | | |
| Recovered | 1223[89.3] | 2994[90.5] | 0.87[0.71–1.08] |
| Referred | 29[2.2] | 90[2.7] | 0.77[0.51–1.18] |
| Deaths | 117[8.5] | 223[6.7] | 1.29[1.02–1.63]* |

Odds ratio for categoric al variables; Mean difference for continuous variables; CI confidence intervals

***p<0.001

**p<0.01

*p<0.05.

did not inquire the type of anti-hypertensive patients in our study cohort. Certain BP medications such as renin-angiotensin system blockers are known to be useful in COVID-19 [20, 21].

Previous meta-analyses including studies from India have identified diabetes as equally important as hypertension for adverse COVID-19 related outcomes [11, 12]. In the present study the unadjusted OR for diabetes and deaths were 1.88 (CI 1.46–2.43), however, the risk significantly attenuated after age and sex adjustment to OR 1.16 (CI 0.89–1.52) which is

**Table 6. Univariate and multivariate logistic regression analyses (odds ratio and 95% confidence intervals) for in-hospital deaths in various cardiovascular risk groups.**

| Risk factor variables | Unadjusted Odds Ratios | Adjusted for age and sex | Adjusted for age, sex, household size, risk factors, comorbidities |
|---|---|---|---|
| Age ≥60 yr vs <40 yr | 8.47[5.87–12.21] | 8.49[5.88–12.25]* | 7.25[4.92–10.66]* |
| Age 40–59 yr vs <40y | 3.69[2.53–5.38] | 3.66[2.50–5.33]* | 3.08[2.08–4.56]* |
| Men vs Women | 1.88[1.41–2.51] | 1.26[0.91–1.48]** | 1.05[0.82–1.35]** |
| Hypertension | 2.22[1.74–2.83] | 1.32[1.02–1.70] | 1.27[0.94–1.73] |
| Diabetes | 1.88[1.46–2.43] | 1.16[0.89–1.52] | 1.02[0.75–1.40] |
| Tobacco | 1.29[1.02–1.63] | 1.28[1.00–1.63] | 1.23[0.96–1.57] |

*sex-adjusted

** age adjusted.

**Table 7. Univariate and multivariate logistic regression analyses (odds ratio and 95% confidence intervals) for secondary outcomes in various cardiovascular risk factor groups.**

| | Risk factor variables | Unadjusted Odds Ratios | Adjusted for age and sex | Adjusted for age, sex, household size, risk factors, comorbidities |
|---|---|---|---|---|
| Invasive ventilation | Age ≥60 yr vs <40 yr | 4.94[3.10–7.86]* | 5.02[3.15–7.99]* | 3.98[2.36–6.71] |
| | Age 40–59 yr vs <40y | 2.97[1.86–4.77]* | 2.96[1.85–4.75] | 2.67[1.60–4.43] |
| | Men vs Women | 1.00[0.71–1.42]** | 0.94[0.67–1.33]** | 1.09[0.76–1.55] |
| | Hypertension | 3.05[2.22–4.20] | 2.12[1.51–2.97] | 2.64[1.78–3.92] |
| | Diabetes | 1.64[1.14–2.35] | 1.12[0.77–1.62] | 0.64[0.42–0.99] |
| | Tobacco | 1.27[0.92–1.76] | 1.24[0.89–1.72] | 1.12[0.79–1.57] |
| Non-invasive ventilation | Age ≥60 yr vs <40 yr | 3.21[2.37–4.35] | 3.17[2.33]4.29]* | 2.85[1.77–3.50]* |
| | Age 40–59 yr vs <40y | 2.17[1.60–2.94] | 2.16[1.59–2.93]* | 1.99{1.44–2.74}* |
| | Men vs Women | 0.88[0.69–1.33] | 0.93[0.73–1.19]** | 0.92[0.65–1.31]** |
| | Hypertension | 1.82[1.41–2.35] | 1.29[0.99–1.69] | 0.91[0.63–1.32] |
| | Diabetes | 1.21[0.91–1.60] | 0.86[0.64–1.15] | 0.49[0.34–0.69] |
| | Tobacco | 1.31[1.04–1.66] | 1.32[1.04–1.67] | 0.89[0.69–1.14] |

*sex-adjusted

** age adjusted.

different from the previous studies. In the present study, we included patients with known diabetes only and this is a study limitation [22]. It is likely that using biomarkers for diabetes diagnosis (HbA1c, glucose tolerance test, etc.) we would have diagnosed more individuals with diabetes, but these criteria are fraught with inconsistency during any acute illness. We also found significant association of smoking/tobacco use with death and other adverse outcomes in our cohort. This association holds even after multivariate adjustments and shows that

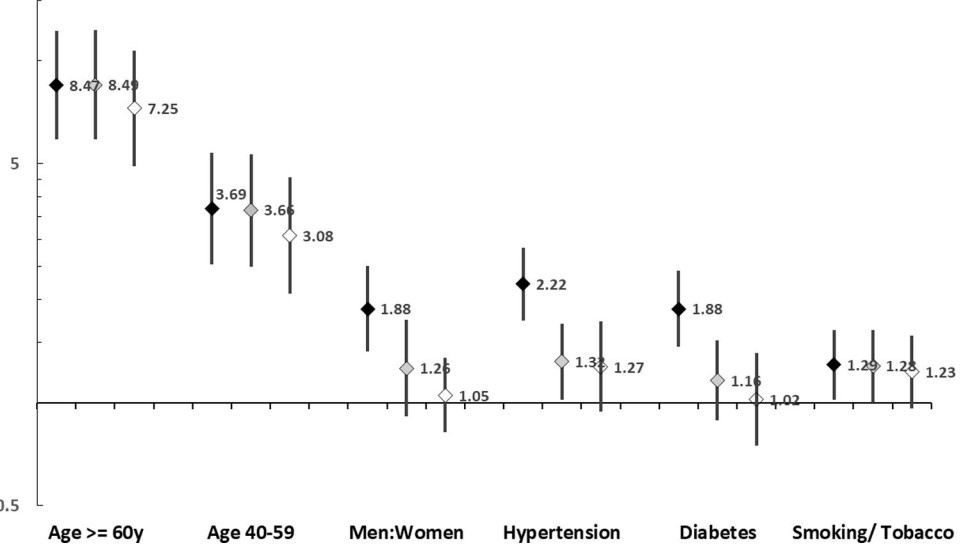

**Fig 3. Odds ratio and 95% confidence intervals for COVID-19 related deaths in patients in old vs young, men vs women, hypertension, diabetes and tobacco groups on univariate (black markers), age and sex adjusted (grey markers) and multivariate (open markers) logistic regression.**

tobacco is an important risk factor. This is different from some previous studies that have reported disparate results [23]. On the other hand, a large meta-analysis that included 109 studies with 517,020 patients reported that smoking was associated with increased risk of admission to ICU and increased mortality (OR 1.58, CI 1.38–1.81) [24]. Meta-regression analyses identified that the increased risk of smoking was mediated via increased age, hypertension and diabetes. Our finding is similar to data from Chinese cohorts (high rates of smoking) in the aforementioned meta-analysis [24].

The study has several other limitations. This is a single-centre study and the results may not be externally valid within India or other countries as Rajasthan is one of the less-developed states in the nation with lower prevalence of hypertension and diabetes [19, 25]. On the other hand, this is the largest study from India and much larger than many other studies from developed countries, the data were obtained from a government hospital thus assuring wider population representation and better data granularity. Secondly, this is not a population-based study as many studies from Europe and North America [7], and we may have missed data on milder forms of disease. Thirdly, we do not have data on obesity or body-mass index which is an important COVID-19 risk factor in hospital- and population-based studies [7, 24]. Fourthly, we also do not have data on biochemical investigations for all the patients, although data on white cell count are available for more than 90%. Also, we do not have data of radiological evaluation of all the patients as it is well known that computerized tomographic images provide important prognostic information [26]. Fifthly, the rate of progression of illness as well as greater details of causes of deaths are not available and this is a study limitation as discussed earlier [17]. We did not analyze data on patients with known cardiovascular disease, chronic respiratory disease, cancers and chronic kidney disease because of small numbers of these patients (Fig 1). And finally, we did not obtain data regarding post-COVID syndrome which is emerging as important health problem especially in persons with comorbidities [27]. On the other hand, this is the largest study from India and with robust data has important clinical implications, especially in view of the ongoing third wave in the country [8].

In conclusion, this study shows that older patients, males, and those with hypertension, diabetes and any tobacco use have greater risk of death and adverse outcomes from COVID-19. Attenuation of risk with age-adjustment shows that increasing age is the most important factor of risk. It is recommended that individuals with cardiovascular risk factors, especially older men and women, should be focus of public health measures and must be informed regarding increased risk of death in COVID-19. Moreover, these high risk individuals must aggressively follow all non-pharmacological physical measures for prevention [28]. These groups should also be prioritized for primary vaccinations and vaccine-boosters [29]. Clinicians are advised to seek early evidence of deterioration of pulmonary function and signs of cardiovascular and extrapulmonary manifestation of acute COVID-19 in these patients and provide optimum management [30, 31]. It is likely that with proper preventive and therapeutic interventions the higher risk of adverse outcomes in COVIDF-19 patients with cardiovascular risk factors can be mitigated.

## Supporting information

**S1 Checklist. STROBE checklist.**
(DOCX)

## Author Contributions

**Conceptualization:** Arvind K. Sharma, Vaseem Naheed Baig, Raja Babu Panwar, Vishwa Mohan Katoch, Rajeev Gupta.

**Data curation:** Arvind K. Sharma, Sonali Sharma.

**Formal analysis:** Arvind K. Sharma, Sonali Sharma, Rajeev Gupta.

**Investigation:** Sonali Sharma, Gaurav Dalela.

**Methodology:** Arvind K. Sharma, Gaurav Dalela.

**Project administration:** Arvind K. Sharma, Vaseem Naheed Baig, Rajeev Gupta.

**Software:** Arvind K. Sharma.

**Supervision:** Arvind K. Sharma, Vaseem Naheed Baig, Sonali Sharma, Gaurav Dalela, Raja Babu Panwar, Vishwa Mohan Katoch, Rajeev Gupta.

**Validation:** Arvind K. Sharma, Sonali Sharma.

**Writing – original draft:** Arvind K. Sharma, Rajeev Gupta.

**Writing – review & editing:** Arvind K. Sharma, Vaseem Naheed Baig, Sonali Sharma, Gaurav Dalela, Raja Babu Panwar, Vishwa Mohan Katoch, Rajeev Gupta.

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
