## [Decision Letter · Decision Letter 0]

11 Jan 2022

PGPH-D-21-01016

Cardiovascular Risk Factors and Outcomes in COVID-19: Hospital-Based Prospective Study in India

Dear Dr. Gupta,

Thank you for submitting your manuscript to PLOS Global Public Health. After careful consideration, we feel that it has merit but does not fully meet PLOS Global Public Health’s publication criteria as it currently stands. Therefore, we invite you to submit a revised version of the manuscript that addresses the points raised during the review process.

We look forward to receiving your revised manuscript.

Kind regards,

Manish Barman, MD., MSc., FRCP

Academic Editor

Journal Requirements:

1. Please provide separate figure files in .tif or .eps format only, and remove any figures embedded in your manuscript file.  If you are using LaTeX, you do not need to remove embedded figures.

For more information about figure files please see our guidelines: https://journals.plos.org/globalpublichealth/s/figures

Additional Editor Comments (if provided):

Dear Authors

I agree with the suggestions and inputs from the reviewers. Kindly incorporate these suggestions.

Thanks

Reviewers' comments:

Reviewer's Responses to Questions

**Comments to the Author**

1. Does this manuscript meet PLOS Global Public Health’s publication criteria? Is the manuscript technically sound, and do the data support the conclusions? The manuscript must describe methodologically and ethically rigorous research with conclusions that are appropriately drawn based on the data presented.

Reviewer #1: Partly

Reviewer #2: Yes

2. Has the statistical analysis been performed appropriately and rigorously?

Reviewer #1: No

Reviewer #2: Yes

3. Have the authors made all data underlying the findings in their manuscript fully available (please refer to the Data Availability Statement at the start of the manuscript PDF file)?

Reviewer #1: Yes

Reviewer #2: Yes

4. Is the manuscript presented in an intelligible fashion and written in standard English?

Reviewer #1: Yes

Reviewer #2: Yes

5. Review Comments to the Author

Reviewer #1: This is a large single-centre study exploring predictors COVID-19 outcomes in hospitalised patients. Whilst many such studies have already been published, this is the largest such study from India and likely one of the largest from a LMIC.

The specific aim was to explore the association of cardiovascular risk factors with COVID-19 outcomes. The rationale for restricting consideration to cardiovascular risk factors only is unclear. Why not consider all the variables that the authors collected data for - for example they have data on pulmonary disease? Also, how was the cut-off for age selected? A binary cut-off of 50 years greatly limits the information we can take from the analysis. Age could either be used as a continuous variable, or could be categorised in to multiple levels (e.g. <40, 49-59, 60-69, 70+). For Table 6 it would be better to present the 'full' adjusted models as separate tables - i.e. one table per model showing all the factors adjusted for - this would make it much easier to interpret.

There is some lack of clarity around whether the authors were aiming to describe outcomes in these groups (i.e. based on unadjusted analysis only) or to determine whether these are independent risk factors (i.e. based on adjusted analysis). A factor may have a significant association with an outcome on an unadjusted analysis, but after adjusting for possible confounders there is no longer a significant association (i.e. patients with this factor have higher risk of the outcome, but it isn't actually because of this factor but due to some other confounder). This is largely what has happened in this study. Whilst all the CV variables are associated with mortality, in table 6 only age is an independent predictor of mortality. Similarly, only hypertension is an independent predictor of invasive ventilation and only age is a independent predictor of non-invasive ventilation. Seeing the full models would help with interpretation of this. So although the authors state that "this study shows that older patients, males, and those with hypertension, diabetes and any tobacco use have greater risk of deaths and adverse outcomes from COVID-19" it is not true that all these factors are independently associated with those outcomes.

Other comments

1. Please make use of a relevant reporting guideline and complete the appropriate checklist, e.g. STROBE

2. Please define a single primary outcome and secondary outcomes

3. The authors describe this as a prospective study. Does this mean that patients were prospectively identified and outcomes collected at the point of hospital discharge?

4. Were children included - this is not mentioned

Dmitri Nepogodiev

University of Birmingham, UK

Reviewer #2: The study methodology is not quite clear although the author indicated a prospective study in the title; it was not appreciated in the methodology. For a prospective study, features such as duration of the study and time of outcomes should be specified which was not done, author only stated the period for enrollment. If indeed a prospective design was employed, a parameter of interest to the author such as BMI could have been measured.

I think the study used an existing database of patients containing registry of case files and medical history hence reporting on suspected Covid-19 cases was not necessary since the study sought to look for outcomes in Covid-19 and not outcomes in suspected Covid-19. The author should give information on Covid-19 (confirmed by RT-PCR) since it was readily available in the hospital registry.

Finally, no sample size was calculated for the study so I assumed every unit of the population was captured. The author should be more modest about presentation of his methodology, apart from these observations the study is good.

6. PLOS authors have the option to publish the peer review history of their article (what does this mean?). If published, this will include your full peer review and any attached files.

**Do you want your identity to be public for this peer review?** For information about this choice, including consent withdrawal, please see our Privacy Policy.

Reviewer #1: No

Reviewer #2: **Yes: **Kwakye George Kumi

---

## [Editor Report · Decision Letter 1]

1 Feb 2022

Cardiovascular Risk Factors and Outcomes in COVID-19: A Hospital-Based Study in India

PGPH-D-21-01016R1

Dear Dr. Gupta,

We are pleased to inform you that your manuscript 'Cardiovascular Risk Factors and Outcomes in COVID-19: A Hospital-Based Study in India' has been provisionally accepted for publication in PLOS Global Public Health.

Best regards,

Manish Barman, MD., MSc., FRCP

Academic Editor

Dear Authors

Happy to note that you have accepted the major suggestions and incorporated them in your revision.

I have made my decision and forwarded it to the staff Editor.

Best Regards

Academic Editor